# Associations between Florida counties' COVID-19 case and death rates and meaningful use among Medicaid providers: Cross-sectional ecologic study

Katherine Freeman[1]⊙ *, Judith P. Monestime[2]⊙

**1** Division of Biomedical Sciences, Charles E. Schmidt College of Medicine, Florida Atlantic University, Boca Raton, Florida, United States of America, **2** Health Administration Programs, Management Department, College of Business, Florida Atlantic University, Boca Raton, Florida, United States of America

⊙ These authors contributed equally to this work.
* kfreemancostin@health.fau.edu

**Data Availability Statement:** All data are in the manuscript and supporting information files.

**Funding:** The authors received no specific funding for this work.

## Abstract

Although the Health Information Technology for Economic and Clinical Health (HITECH) Act has accelerated adoption of Electronic Health Records (EHRs) among Medicaid providers, only half achieved Meaningful Use. Furthermore, Meaningful Use' impact on reporting and/or clinical outcomes remains unknown. To address this deficit, we assessed the difference between Medicaid providers who did and did not achieve Meaningful Use regarding Florida county-level cumulative COVID-19 death, case and case fatality rates (CFR), accounting for county-level demographics, socioeconomic and clinical markers, and healthcare environment. We found that cumulative incidence rates of COVID-19 deaths and CFRs were significantly different between the 5025 Medicaid providers not achieving Meaningful Use and the 3723 achieving Meaningful Use (mean 0.8334/1000 population; SD = 0.3489 vs. mean = 0.8216/1000; SD = 0.3227, respectively) (P = .01). CFRs were .01797 and .01781, respectively, P = .04. County-level characteristics independently associated with increased COVID-19 death rates and CFRs include greater concentrations of persons of African American or Black race, lower median household income, higher unemployment, and higher concentrations of those living in poverty and without health insurance (all P < .001). In accordance with other studies, social determinants of health were independently associated with clinical outcomes. Our findings also suggest that the association between Florida counties' public health outcomes and Meaningful Use achievement may have had less to do with using EHRs for reporting of clinical outcomes and more to do with using EHRs for coordination of care—a key measure of quality. The Florida Medicaid Promoting Interoperability Program which incentivized Medicaid providers towards achieving Meaningful Use, has demonstrated success regarding both rates of adoption and clinical outcomes. Because the Program ends in 2021, we support programs such as HealthyPeople 2030 Health IT which address the remaining half of Florida Medicaid providers who have not yet achieved Meaningful Use.

**Competing interests:** The authors have declared that no competing interests exist.

## Author summary

Our study investigates whether government support during 2012–2018 to increase technology among Medicaid providers to implement electronic health records is associated with reported COVID-19 death and case rates. We found that reported cumulative COVID-19 death rates were significantly greater in counties where more providers lacked the technology. Case rates were not associated with providers' access to this technology. Counties with greater percentages of residents who are African American/Black, unemployed, living in poverty, underinsured and with histories of chronic lower respiratory disease, influenza or pneumonia were associated with worse clinical outcomes. Our findings also suggest that this technology may have been used primarily to coordinate patient care—a key quality indicator, rather than for reporting outcomes to public health agencies. Because program year 2021 marked the end of the Florida Promoting Interoperability Incentive Program, our findings provide baseline data to extend this support beyond 2021 in alignment with the Healthy People 2030 health information technology objectives. Increasing the proportion of office based Medicaid providers using advanced EHRs functions will help minimize an emerging "digital divide" between the quality of care provided to the nation's most vulnerable patients who access care from Medicaid providers not having achieved meaningful use vs. those Medicaid providers using advanced EHR functions.

## Introduction

### Background

Accurate reporting to public health agencies has never been more critical than during the COVID-19 pandemic. Communication of daily incidence rates and trends in COVID-19 cases and deaths promotes optimal utilization of existing resources by healthcare professionals and helps mitigate population risk by identifying communities with increased spread while emphasizing protective measures [1]. Quantification of these key epidemiologic markers provides evidence-based decision-making regarding economic recovery, school participation, and healthcare resource allocation for staffing, testing, and vaccine distribution [2]. The primary reporting mechanism to state, territorial and local public health agencies was revolutionized by implementing electronic health records (EHR) which improve efficiency and quality of care in the healthcare system by facilitating public health reporting and surveillance of infectious disease transmission [1, 3]. Public health agencies have found EHRs particularly useful in surveillance efforts to curtail bioterrorism and outbreaks [4]. Not only does linkage of patient health records among providers facilitate coordination of care [5] and promote more optimal clinical outcomes, [6] but it allows public health administrators to initiate custom queries within secure data environments [4]. Additionally, the use of EHRs has fostered change in health record formats toward standardized reporting. The Health Information Technology for Economic and Clinical Health (HITECH) Act was enacted in February 2009; its purpose was to encourage healthcare providers to convert paper records to electronic and utilize such technology in accordance with federal regulations, commonly referred to as Meaningful Use, to improve quality-of-care outcomes, promote public health reporting and reduce health disparities [7, 8].

The Promoting Interoperability Program has been operational since 2011 and is currently providing incentive payments to Eligible Professionals (EPs) for demonstrating Meaningful Use [9, 10]. The HITECH Act provided subsidy payments [11], about $21,250 during year 01

and $8500 each subsequent year, to Medicaid providers who were required to meet standards of advanced functions, thus achieving Meaningful Use [7, 9, 12]. HITECH's overall goal was to improve population health outcomes through the timely and seamless transfer of health data [8, 13–17]. Despite the promising benefits of public health reporting, in 2014, only 6% of providers chose Meaningful Use objectives to submit syndromic data to public health agencies [18]. Furthermore, of the $6 billion investment to Medicaid providers nationally, only 56.2% had demonstrated Meaningful Use by 2018 [19]. Ultimately, expected gains from the Triple Aim to improve population health outcomes, part of the impetus behind public health reporting requirements for Meaningful Use, has lagged [20].

Significant investments through the HITECH Act, for the Medicaid Promoting Interoperability program, to implement EHRs and achieve Meaningful Use has greatly contributed to the uptake of a basic technology in clinical care settings [21]. The use of EHRs for public health surveillance and reporting holds promise to identity cases that meet certain criteria for automatic reporting to public health agencies [12]. EHRs can also facilitate efforts to respond to public health emergencies such as the COVID-19 pandemic, to bridge the gaps among clinical practices to promote more equitable outcomes. While EHRs by themselves are useful, certain advanced EHR capabilities which are required to achieve Meaningful Use are necessary to exchange health data with public health departments. However, the trend in achieving Meaningful Use has stalled among Medicaid providers, who serve a large population of traditionally underserved and marginalized patients [10, 22]. The Medicaid Meaningful Use rate for Florida providers (42.6%), is more than 13 percentage points below the national provider Medicaid Meaningful Use rate of 56.2% [22, 23]. Ultimately, examining the differences in reported incidence rates of COVID-19 cases and deaths between Medicaid providers who did and did not achieve Meaningful Use, can reveal barriers to public health reporting of COVID-19 outcomes which inform public health decision-making. Furthermore, according to the Commonwealth Fund, in 2018, Florida ranked 48th lowest among 50 states plus the District of Columbia regarding overall healthcare and 49th for access, quality, and use of healthcare [24]. Florida ranks third among states with the highest Medicaid enrollment, behind California and New York [25]. In 2020, one in five Florida residents was on Medicaid, comprising 3,716,747 Medicaid beneficiaries; of these, 27% were Black or African American, 29% Hispanic or LatinX, 34% White, and 11% Other [26]. Additionally, 3 of 7 children in Florida and 4 of 7 of those in nursing homes are covered by Medicaid [26]. At the time of writing, Florida ranked 11th among states with the fewest coronavirus restrictions [27] and was tied for the largest percentage of its population 65 years and older (20%) [28] who are most at risk of death or requiring hospitalization. As previously mentioned, Florida Medicaid Meaningful Use low achievement rate in comparison to the national rate [23], underscores the limited advanced reporting capabilities of most of the state's Medicaid providers.

Due to increased COVID-19 risk among Floridians because of these factors and pronounced health disparities cited during the COVID-19 pandemic at the county, state, and national levels [29], the two-sided hypotheses alternative to the null hypothesis of no difference are: 1) COVID-19 death, case and case fatality rates are underreported (lower) among Medicaid providers not achieving Meaningful Use relative to those achieving Meaningful Use, and, 2) these reported outcomes are greater in counties with lower concentrations of this technology. Although researchers have begun to explore the associations between COVID-19 outcomes and demographic and economic risk factors that relate to health equity, these studies have not emphasized that Health Information Technology (Health IT) is an essential component of Social Determinants of Health (SDoH) [29, 30].

## Objectives

Our research addresses how a possible gap in achievement of advanced EHR functions which is consistent with the Healthy People 2030 initiative for Health Care Access and Quality, is associated with public health reported COVID-19 cases and deaths. According to this initiative, providing office-based physicians with the necessary electronic information at the point of care is a high-priority public health issue. Our premise is that Medicaid providers who did not achieve Meaningful Use with its advanced reporting functions will be associated with lower reported Florida county-level incidence rates of COVID-19 cases and deaths accounting for county-level SDoH, than those Medicaid providers who achieved Meaningful Use [31].

## Methods

### Study design, setting and participants

In this cross-sectional ecologic study, we investigated the association between Meaningful Use achievement and Florida county-level incidence rates of COVID-19 cases and death rates reported on publicly available data sets as of November 19, 2020. The date of the first documented case of COVID-19 in Florida was March 1, 2020. Although we label the design as cross-sectional and report associations, we note that our study focuses on Medicaid providers enrolled in HITECH between 2012 and 2016 with a minimum of two years of follow-up occurring prior to the pandemic, and thus the design has a longitudinal component. Furthermore, because Meaningful Use was captured before the COVID-19 pandemic, case and death rates are not confounded by it. This study was deemed exempt by the university's Institutional Review Board for the Protection of Human Subjects, involving publicly available de-identified data only. We followed the Strengthening the Reporting of Observational Studies in Epidemiology (STROBE) reporting guidelines, with the first table presenting details of each data element, and references to the text [32].

The conceptual model for this study stems from the Agency for Healthcare Research and Quality's (AHRQ) framework for investigating barriers to achieve Meaningful Use among Medicaid providers [33]. The model borrows from previous research in resource dependence theory (RDT), which presumes that the key to organizational success is dependent on the extent to which organizations can acquire and maintain resources [34]. In this context, RDT assumes that Medicaid providers may perceive that HITECH subsidies will assist them in obtaining necessary technical resources for their practice. RDT has been used widely in the healthcare literature, including research on contract management [35], hospital EHRs [36], and most recently, Medicaid provider EHRs [23].

### Measures

**Provider Meaningful Use achievement (Exposure Variable).** Meaningful Use is defined as the Medicaid provider's continuation in the program beyond the first year's incentive payment, documented by the provider having received at least one additional payment during the two years after enrollment. Thus, the observation period for all providers enrolled between 2011 and 2016 was extended through 2018 to allow for a minimum of two years of follow-up to determine Meaningful Use; it was considered unlikely that provider participation would continue after two years of absence in the program [23]. Similar to a previous study, providers were classified by attestation to Meaningful Use by eligibility to receive subsequent payments (Payment Years 2–6) from 2012–2018. We identified the continuation of longitudinal participation in the Florida Medicaid Promoting Interoperability program and provider and practice

characteristics associated with achieving Meaningful Use after receiving the first-year incentive payment [23].

The provider participation database contains the provider's National Provider Identifier (NPI), payment year, program year, provider specialty, EHR Phase, Meaningful Use Stage, and EHR certification number from 2011 to the present. The database includes a unique record for each Medicaid provider, each with a variable indicating whether or not the provider received payments subsequent to their first-year incentive payment, demonstrating Meaningful Use. The county name on the Florida Agency for Healthcare Administration (AHCA) Provider Participation Database was obtained from the National Plan and Provider Enumeration System (NPPES) [37].

**Population health and socioeconomic factors (Potential Confounders).** The Area Health Resources File is a public datafile released annually by the Bureau of Health Workforce of the Health Resources and Services Administration [38]. It includes data on SDoH that were considered important and potential confounders of the relationship between Meaningful Use and COVID-19 outcomes. The Area Health Resources Files release year corresponds to the fiscal year (October 1 to September 30) in which the data were published. The Area Health Resources File includes Healthcare Professions, Health Facilities, Population Characteristics including influenza and chronic respiratory disease considered risks during COVID-19, Economics, Health Professions' Training, Hospital Utilization, Hospital Expenditures, and Environment at the county, state, and national levels 50 data sources, identified by zip code which were aggregated into counties.

**COVID-19 case and death rates (Outcomes).** The Florida Department of Health (FDoH) Open Data provided incidence rates of COVID-19 cases and deaths by county cumulative to 11/19/20 [39]. The FDoH includes data on SDoH, considered important and potential confounders of the association between Meaningful Use and COVID-19 outcomes. United States Census Bureau 2019 provided county population totals and demographic data as of July 1, 2019 (V2019) [28].

## Statistical analysis

We used all records (N = 8748) from available public sources, and thus no power analysis was performed. Records with missing data (n = 311; 3.4%) were similar to the 8748 less 311. The Area Health Resources File with derived counties was then linked by county with FDoH's county level and U.S. Census Bureau data. All county rates were derived by dividing the number in the county population with the characteristic of interest by the population for that county, expressed per 1000.

COVID-19 Death Rates: The distributions across providers within counties for the COVID-19 outcome of death were examined, and the unadjusted means and standard deviations are presented by selected variables. For each continuous variable, the median was used as the cut point for reporting descriptive statistics. One of the pair of potentially confounding variables resulting in bivariate correlations of .4 or greater was removed due to multicollinearity. Provider variables in the initial mixed effects model included: specialty type, Eligible Volume Criteria (qualified based on their own practice alone (solo) or as part of their group practice) and program year; variables for the county population included: race, ethnicity, three-year rate of Influenza and Pneumonia (2015–17), three-year rate of Chronic Lower Respiratory Disease (2015–17), Unemployment Rate for ages 16+ from 2018, Per Capita Personal Income from 2017, Median Household Income from 2017, Percent Persons living in Poverty from 2017, % Persons 65+ in Deep Poverty from 2013–17, and %'s under 19 and 18–64 years without Health Insurance from 2017. Meaningful Use was considered the exposure

variable and retained in all models. At each iteration, the variable with the greatest non-significant p-value ($P> = .05$) was removed, and the model rerun to derive the final parsimonious model that included significant variables only.

COVID-19 Case Rates: the same bivariate and multivariate modeling approach was performed as per the outcome 'death'. Because the difference in case rates between those Medicaid providers who achieved Meaningful Use and those who did not was not significant, interactions were examined in post hoc analyses to assess the significance of effect modifiers, using the same modeling approach as stated previously. We acknowledge that CFR is a function of death rate, and expect results to be similar; however, we use the same modeling approach described above.

All tests of significance were two-tailed and performed at $\alpha = .05$ using SAS Version 9.4, Cary NC.

## Results

### Setting and participants

Florida is a peninsula extending from the southeast region of mainland United States, with more than half (52.3%) of its 67 counties bordering the coast. The following describes characteristics across counties: median population size 132,420, range: 8354 to 2,716,940; median percent of those of African American or Black race 11%, range: 2.7% to 55.3%; median percent of those of Hispanic or LatinX ethnicity 9.6%, range: 2.4% to 68.0%. The median age across county populations was 42.5 years, range: 30.7, 67.0. The median income for Florida in 2019 was $59,227 compared with $68,703 for the U.S [28].

### COVID-19 death rates and CFR

Table 1 presents descriptive statistics for the rate of COVID-19 deaths /1000 population, stratified by characteristics of 8748 eligible Medicaid providers, including setting, enrollment year, and demographic and socioeconomic characteristics of the population at the county level. Variables (i.e., provider's specialty, eligible Medicaid patient volume criteria, EHR vendor, geographic areas, and year of enrollment) were chosen because they encompass SDoH and possibly confound the association between Meaningful Use and COVID-19 outcomes [27]. Means and standard deviations are presented for the raw data, and least-squares means and standard deviations for the multivariate model; bivariate and multivariate p-values for differences for outcomes between subgroups within variables are also presented. Variables in the final multivariate model are those with resulting $P$-values less than .05. From multivariate analyses, COVID-19 death rates were significantly different between providers who achieved Meaningful Use and those who did not ($P = .01$), with relatively more deaths reported among those who did not progress past program year 1. County-level characteristics associated with greater COVID-19 death rates were: higher concentrations of persons of African American or Black race ($P < .001$), higher prevalence rates of chronic lower respiratory disease ($P < .001$), and of influenza and pneumonia ($P < .001$), higher per capita personal income ($P < .001$), lower median household income ($P < .001$), higher unemployment rate ($P < .001$), and higher rates of those living in poverty ($P < .001$) and without health insurance ($P < .001$). County-level characteristics associated with lower COVID-19 death rates were higher concentrations of persons of American Indian/Alaska Native race ($P < .001$) or Hispanic/LatinX ethnicity ($P < .001$), and higher concentration of persons ages 65 years and older living in deep poverty ($P < .001$). Qualifying based on their own practice's eligible volume criteria ($P < .001$) and practicing dentistry ($P = .03$) were associated with increased COVID-19 death rates. Because CFR is a function of both death and case rates, thus not independent of death rate, results are

**Table 1. Characteristics associated with COVID-19 death rates among Florida county populations (per 1000).**

| CHARACTERISTICS | EPs N = 8748 | Mean rate covid deaths /1000[a] | Standard Deviation [a] | Bivariate P -Value[a] | LSM[b] | LSM Standard Error[b] | P-value from Multi-variate model[b] |
|---|---|---|---|---|---|---|---|
| **Meaningful Use** | | | | .77 | | | 0.01 |
| No Meaningful Use | 5025 | 0.8334 | 0.3489 | | 0.8102 | 0.0034 | |
| Achieved Meaningful Use | 3723 | 0.8216 | 0.3227 | | 0.8092 | 0.0039 | |
| **Eligible Volume Criteria[c]** | | | | < .001 | | | < .001 |
| Group | 5612 | 0.8496 | 0.3303 | | 0.7943 | 0.0038 | |
| Solo | 3136 | 0.7904 | 0.3484 | | 0.8251 | 0.0036 | |
| **Dentists[c]** | | | | < .001 | | | 0.03 |
| Yes | 707 | 0.7375 | 0.3162 | | 0.8103 | 0.0050 | |
| No | 8041 | 0.8364 | 0.3388 | | 0.8091 | 0.0029 | |
| **Nurse Practitioners** | | | | < .001 | | | NI[d] |
| Yes | 2064 | 0.7912 | 0.3384 | | | | |
| No | 6684 | 0.8398 | 0.3372 | | | | |
| **Pediatricians** | | | | .24 | | | NI |
| Yes | 1218 | 0.8082 | 0.3200 | | | | |
| No | 7530 | 0.8316 | 0.3408 | | | | |
| **Program Year** | | | | < .001 | | | NI |
| 2011 | 2635 | 0.8661 | 0.3398 | | | | |
| 2012 | 2060 | 0.7902 | 0.3220 | | | | |
| 2013 | 1638 | 0.7975 | 0.3481 | | | | |
| 2014 | 877 | 0.9245 | 0.3499 | | | | |
| 2015 | 793 | 0.7597 | 0.2962 | | | | |
| 2016 | 745 | 0.8282 | 0.3415 | | | | |
| **% non-white in population per 1000** | | | | .73 | | | NI |
| < Median NW | 1999 | 0.7755 | 0.1838 | | | | |
| > = Median NW | 6612 | 0.8402 | 0.3687 | | | | |
| **% African American/Black Population 2010[e]** | | | | < .001 | | | < .001 |
| < Median | 3894 | 0.8001 | 0.2141 | | 0.7674 | 0.0048 | |
| > = Median | 4854 | 0.8510 | 0.4099 | | 0.8521 | 0.0030 | |
| **% American Indian/Alaska Native in Population 2010** | | | | < .001 | | | < .001 |
| < Median | 1745 | 1.3311 | 0.0978 | | 0.8491 | 0.0066 | |
| > = Median | 7003 | 0.7031 | 0.2484 | | 0.7704 | 0.0025 | |
| **% Hispanic/Latino Population 2010[e]** | | | | < .001 | | | < .001 |
| < Median | 4335 | 0.6920 | 0.2621 | | 0.8656 | 0.0038 | |
| > = Median | 4413 | 0.9623 | 0.3505 | | 0.7538 | 0.0049 | |
| **% 3-Yr Chronic Lower Resp Disease 2015–17** | | | | < .001 | | | < .001 |
| < Median | 4115 | 0.6336 | 0.2621 | | 0.7981 | 0.0045 | |
| > = Median | 4615 | 1.0010 | 0.2988 | | 0.8213 | 0.0038 | |
| **% 3-Yr Influenza & Pneumonia 2015–17** | | | | < .001 | | | < .001 |
| < Median | 3903 | 0.6159 | 0.2339 | | 0.7853 | 0.0035 | |
| > = Median | 4534 | 0.9993 | 0.3058 | | 0.8341 | 0.00524 | |
| **% Per Capita Personal Income 2017** | | | | < .001 | | | < .001 |
| < Median | 4102 | 0.6304 | 0.2654 | | 0.6375 | 0.0059 | |
| > = Median | 4646 | 1.0031 | 0.2964 | | 0.9819 | 0.0031 | |
| **Median Household Income 2017** | | | | < .001 | | | < .001 |

*(Continued)*

**Table 1.** (Continued)

| CHARACTERISTICS | EPs N = 8748 | Mean rate covid deaths /1000[a] | Standard Deviation [a] | Bivariate P -Value[a] | LSM[b] | LSM Standard Error[b] | P-value from Multi-variate model[b] |
|---|---|---|---|---|---|---|---|
| < Median | 4039 | 1.0260 | 0.3535 | | 0.8806 | 0.0039 | |
| > = Median | 4709 | 0.6588 | 0.2071 | | 0.7389 | 0.0040 | |
| **% Unemployment Rate, ages 16 years and older 2018** | | | | < .001 | | | < .001 |
| < Median | 4209 | 0.6142 | 0.2036 | | 0.6684 | 0.0051 | |
| > = Median | 4539 | 1.0270 | 0.3160 | | 0.9511 | 0.0029 | |
| **% Persons ages 65 years & older in Deep Poverty 2013–17[e]** | | | | < .001 | | | < .001 |
| < Median | 3452 | 0.7274 | 0.3088 | | 0.8349 | 0.0045 | |
| > = Median | 5296 | 0.8942 | 0.3401 | | 0.7845 | 0.0034 | |
| **% Persons in Poverty 2017[e]** | | | | .01 | | | < .001 |
| < Median | 3927 | 0.7828 | 0.1835 | | 0.7984 | 0.0045 | |
| > = Median | 4821 | 0.8655 | 0.4206 | | 0.8210 | 0.0032 | |
| **% ages <19 years without Health Insurance 2017** | | | | < .001 | | | < .001 |
| < Median | 3350 | 0.6187 | 0.2137 | | 0.7531 | 0.0044 | |
| > = Median | 5398 | 0.9585 | 0.3356 | | 0.8664 | 0.0039 | |
| **% ages 18–64 years without Health Insurance 2017** | | | | < .001 | | | < .001 |
| < Median | 3849 | 0.6255 | 0.2177 | | 0.7636 | 0.0040 | |
| > = Median | 4899 | 0.9878 | 0.3302 | | 0.8558 | 0.0041 | |

[a] From bivariate analyses.

[b] Estimated Least Squares Means (LSM): LSM are means derived from the mixed effects model that account for other variables in the model; those reported were significant from this multivariate model; characteristics presented were either intrisically dichotomous or continuous variables dichotomized at the median ($P < .05$).

[c] Eligible volume criteria designated as 'solo' means that the percent of their own practice's Medicaid patients qualified the provide to enroll in the program; if the provider did not have sufficient Medicaid patients and if the group in which they practiced did, the provider could enroll designated as 'group'.

[d] NI indicates covariate was not included in the mixed effects initial model because the bivariate p-value >.20 or due to multicolinearity.

[e] Direction of the effect changed from bivariate to multivariate analyses.

EP: eligible providers in the stratification groups within counties.

NOTE: %'s for persons who are of races other than white in the population per 1000 are presented for descriptive purposes only; racial subgroups were used instead in the multivariate model.

NOTE: 311 observations were not used due to missing values, which were similar to those included regarding available data.

NOTE: both per capita income and median household income were included in the initial model given that their bivariate correlation was -0.00777 ($P = .47$).

S1 Data: supporting raw datafile: S1_Data.xlsx

similar to those for death rates. The following summarizes CFR results of significant covariates in the final multivariate model ($P < .001$): African American/Black, American Indian/Alaska Native and Hispanic/Latino; chronic lower respiratory disease; per capita personal income; median household income; living in poverty; without health insurance (ages 18–64); unemployment rate; not achieving Meaningful Use was associated with higher CFRs relative to those achieving Meaningful Use (.01797 and .01781, respectively; $P = .04$).

## COVID-19 case rates

Descriptive statistics for COVID-19 case rates are presented in Table 2. From multivariate analyses, Meaningful Use was not significantly and independently associated with the concentration of COVID-19 cases. Practicing in counties with higher concentrations of persons of

**Table 2. Characteristics associated with COVID-19 case rates among Florida county populations.**

| CHARACTERISTICS | EPs N = 8748 | Mean rate COVID-19 cases /1000[a] | Standard Deviation[a] | Bivariate P-value[a] | LSM[b] | LSM Standard Error[b] | P-value from Multi-variate model[b] |
|---|---|---|---|---|---|---|---|
| **Meaningful Use** | | | | < .001 | | | 0.43 |
| No Meaningful Use | 5025 | 47.1920 | 16.7259 | | 48.4402 | 0.1514 | |
| Achieved Meaningful Use | 3723 | 45.1276 | 15.4170 | | 48.4925 | 0.1641 | |
| **Eligible Volume Criteria[c]** | | | | .29 | | | NI[d] |
| Group | 5612 | 46.3280 | 16.2855 | | | | |
| Solo | 3136 | 46.2872 | 16.0851 | | | | |
| **Dentists** | | | | .17 | | | < .001 |
| Yes | 707 | 44.5512 | 14.1597 | | 48.6331 | 0.2099 | |
| No | 8041 | 46.4683 | 16.3730 | | 48.2997 | 0.1266 | |
| **Nurse Practitioners** | | | | < .001 | | | NI |
| Yes | 2064 | 44.2658 | 15.7465 | | | | |
| No | 6684 | 46.9457 | 16.3037 | | | | |
| **Pediatricians** | | | | < .001 | | | < .001 |
| Yes | 1218 | 44.2428 | 15.3533 | | 48.0528 | 0.1854 | |
| No | 7530 | 46.6483 | 16.3242 | | 48.8800 | 0.1405 | |
| **Program Year** | | | | < .001 | | | NI |
| 2011 | 2635 | 47.2621 | 17.1145 | | | | |
| 2012 | 2060 | 44.3508 | 14.2582 | | | | |
| 2013 | 1638 | 46.4656 | 16.0710 | | | | |
| 2014 | 877 | 51.1656 | 17.7466 | | | | |
| 2015 | 793 | 42.8810 | 14.6346 | | | | |
| 2016 | 745 | 45.9918 | 16.4135 | | | | |
| **% non-white in population per 1000** | | | | < .001 | | | NI |
| < Median | 1999 | 31.3487 | 7.9043 | | | | |
| > = Median | 6612 | 50.5785 | 15.1610 | | | | |
| **% African American/Black Population 2010** | | | | < .001 | | | < .001 |
| < Median | 3894 | 35.1439 | 8.8480 | | 41.0029 | 0.2017 | |
| > = Median | 4854 | 55.2738 | 15.1845 | | 55.9299 | 0.1316 | |
| **% American Indian /Alaska Native Population 2010** | | | | < .001 | | | .003 |
| < Median | 1745 | 73.1197 | 9.3836 | | 54.4484 | 0.2715 | |
| > = Median | 7003 | 39.6338 | 9.0960 | | 42.4843 | 0.1124 | |
| **% Hispanic/Latino Population 2010** | | | | < .001 | | | < .001 |
| < Median | 4335 | 38.3207 | 10.6911 | | 44.8272 | 0.1604 | |
| > = Median | 4413 | 54.1648 | 16.86520 | | 52.1055 | 0.2029 | |
| **% 3-Yr Chronic Lower Respiratory Disease 2015–17** | | | | < .001 | | | < .001 |
| < Median | 4115 | 41.0310 | 8.7165 | | 49.7143 | 0.1887 | |
| > = Median | 4615 | 50.8858 | 19.3131 | | 47.2184 | 0.1602 | |
| **% 3-Yr Influenza & Pneumonia 2015–17** | | | | < .001 | | | < .001 |
| < Median | 3903 | 37.8327 | 8.6586 | | 47.5816 | 0.1485 | |
| > = Median | 4534 | 52.8985 | 17.6331 | | 49.3512 | 0.2166 | |
| **% Per Capita Personal Income 2017** | | | | < .001 | | | < .001 |
| < Median | 4102 | 40.3485 | 10.3178 | | 42.4545 | 0.2415 | |
| > = Median | 4646 | 51.5798 | 18.4891 | | 54.4783 | 0.1366 | |

*(Continued)*

**Table 2.** (Continued)

| CHARACTERISTICS | EPs N = 8748 | Mean rate COVID-19 cases /1000[a] | Standard Deviation[a] | Bivariate P-value[a] | LSM[b] | LSM Standard Error[b] | P-value from Multi-variate model[b] |
|---|---|---|---|---|---|---|---|
| **Median Household Income 2017** | | | | < .001 | | | < .001 |
| < Median | 4039 | 53.4555 | 20.4866 | | 49.7239 | 0.1650 | |
| > = Median | 4709 | 40.1875 | 6.8633 | | 47.2088 | 0.1674 | |
| **% Persons ages 65+ in Deep Poverty 2013–17** | | | | < .001 | | | < .001 |
| < Median | 3452 | 39.7096 | 11.1027 | | 50.2834 | 0.1870 | |
| > = Median | 5296 | 50.6179 | 17.5194 | | 46.6493 | 0.1463 | |
| **% Persons in Poverty 2017** | | | | < .001 | | | < .001 |
| < Median | 3927 | 36.5974 | 8.3871 | | 45.3590 | 0.1905 | |
| > = Median | 4821 | 54.2277 | 16.7382 | | 51.5737 | 0.1400 | |
| **% ages <19 without Health Insurance 2017** | | | | < .001 | | | < .001 |
| < Median | 3350 | 39.2941 | 9.0969 | | 50.7758 | 0.1846 | |
| > = Median | 5398 | 50.6695 | 18.0309 | | 46.1569 | 0.1652 | |
| **% ages 18–64 without Health Insurance 2017** | | | | < .001 | | | < .001 |
| < Median | 3849 | 39.4901 | 9.1792 | | 45.6633 | 0.2128 | |
| > = Median | 4899 | 51.6743 | 18.3821 | | 51.2694 | 0.1303 | |
| **% Unemployment Rate, ages 16 or older 2018** | | | | < .001 | | | < .001 |
| < Median | 4209 | 40.5854 | 7.8550 | | 45.6633 | 0.2128 | |
| > = Median | 4539 | 51.6249 | 19.7688 | | 51.2694 | 0.1303 | |

[a] From bivariate analyses.

[b] Estimated Least Squares Means (LSM): LSM are means derived from the mixed effects model that account for other variables in the model; those reported were significant from this multivariate model; characteristics presented were either intrisically dichotomous or continuous variables dichotomized at the median ($P < .05$).

[c] Eligible volume criteria designated as 'solo' means that the percent of their own practice's Medicaid patients qualified the provide to enroll in the program; if the provider did not have sufficient Medicaid patients and if the group in which they practiced did, the provider could enroll designated as 'group'.

[d] NI indicates covariate was not included in the mixed effects initial model because the bivariate p-value >.20 or due to multicolinearity.

EP: eligible providers in the stratification groups within counties.

NOTE: %'s for persons who are of races other than white in the population per 1000 are presented for descriptive purposes only; racial subgroups were used instead in the multivariate model.

NOTE: 311 observations were not used due to missing values.

S1 Data: supporting raw datafile: S1_Data.xlsx

African American or Black race ($P < .001$) and those of Hispanic/Latino ethnicity ($P < .001$) was associated with higher COVID-19 case rates. Practicing in counties with a higher prevalence of influenza and pneumonia ($P < .001$) was associated with higher COVID-19 case rates. Lower median household income ($P < .001$), and higher rates of poverty ($P < .001$), unemployment ($P < .001$), and adults without health insurance ($P < .001$) were associated with higher COVID-19 case rates. Characteristics associated with lower COVID-19 case rates were practicing pediatrics ($P < .001$), higher prevalence rates of chronic lower respiratory disease ($P < .001$), percent of persons of American Indian/Alaska Native race ($P < .01$), and prevalence of persons ages 65 and over living in deep poverty ($P < .001$). Eligible volume criteria, practicing as a nurse practitioner and program year were not significantly associated with COVID-19 case rates.

Further exploratory (secondary) analyses were conducted to determine whether the association between Meaningful Use and COVID-19 case rates is modified by those county and

**Table 3. Meaningful Use significant interactions with characteristics as they relate to the oncentration of COVID-19 cases / 1000 population.**

| CHARACTERISTICS[a] | N | Mean rate COVID-19 cases /1000[a] | Standard Deviation[a] | P-value for interaction[a] |
|---|---|---|---|---|
| **Dentists** | | | | < .001 |
| Yes * No Meaningful Use | 652 | 44.9331 | 14.3449 | |
| No * No Meaningful Use | 4373 | 47.5287 | 17.0282 | |
| Yes * Achieved Meaningful Use | 55 | 40.0235 | 10.8589 | |
| No * Achieved Meaningful Use | 3668 | 45.2041 | 15.4634 | |
| **% 3-Yr Chronic Lower Respiratory Disease (CLRD) 2015–17** | | | | 0.002 |
| < Median CLRD * Not Meaningful Use | 2675 | 52.5225 | 19.8877 | |
| > = Median CLRD * No Meaningful Use | 2342 | 41.0327 | 8.5590 | |
| < Median CLRD * Achieved Meaningful Use | 1940 | 48.6289 | 18.2573 | |
| > = Median CLRD * Achieved Meaningful Use | 1773 | 41.0288 | 8.9227 | |
| **% 3-Yr Influenza & Pneumonia (Flu) 2015–17** | | | | < .001 |
| < Median Flu * No Meaningful Use | 2631 | 54.3580 | 18.2683 | |
| > = Median Flu * No Meaningful Use | 2267 | 38.3595 | 8.8680 | |
| < Median Flu * Achieved Meaningful Use | 1903 | 50.8807 | 16.5087 | |
| > = Median Flu * Achieved Meaningful Use | 1636 | 37.1028 | 8.3073 | |
| **% Per Capita Personal Income (PI) 2017** | | | | < .001 |
| < Median PI * No Meaningful Use | 2697 | 53.0780 | 19.1316 | |
| > = Median PI * No Meaningful Use | 2328 | 40.3730 | 9.6571 | |
| < Median PI * Achieved Meaningful Use | 1949 | 49.5068 | 17.3539 | |
| > = Median PI* Achieved Meaningful Use | 1774 | 40.3164 | 11.1283 | |
| **% Persons ages 65+ in Deep Poverty (DP) 2013–17** | | | | .02 |
| < Median DP* No Meaningful Use | 3038 | 52.0717 | 18.2283 | |
| > = Median DP* No Meaningful Use | 1987 | 39.7311 | 10.3668 | |
| < Median DP* Achieved Meaningful Use | 2258 | 48.6618 | 16.3185 | |
| > = Median DP* Achieved Meaningful Use | 1465 | 39.6803 | 12.0327 | |
| **% Unemployment Rate (UR) ages 16 or older 2018** | | | | .02 |
| < Median UR* No Meaningful Use | 2601 | 53.3562 | 20.2878 | |
| > = Median UR* No Meaningful Use | 2424 | 40.5776 | 7.3376 | |
| < Median UR* Achieved Meaningful Use | 1938 | 49.3013 | 18.8062 | |
| > = Median UR* Achieved Meaningful Use | 1785 | 40.5961 | 8.5095 | |

[a] Characteristics with significant interactions ($P < .05$) with Achieved vs. Not Achieved Meaningful Use from final mixed effects regression model that included significant main effects and interactions; descriptive statistics are from unadjusted bivariate associations.

NOTE: the symbol '*' denotes 'interaction with'.

S1 Data: supporting raw datafile: S1_Data.xlsx

S1 Text: output for interaction between age and deep poverty: S1_Text.doc

provider characteristics shown in Table 2; results of significant interactions are shown in Table 3. For counties with lower rates of chronic respiratory disease, and of influenza and pneumonia, COVID-19 case rates were significantly lower among providers who achieved Meaningful Use ($P = .002$ and $P < .001$, respectively); for counties with higher rates of disease and illness, the difference in case rates due to achieving Meaningful Use was not pronounced. Among counties with lower per capita incomes, providers who achieved Meaningful Use had significantly lower COVID-19 case rates than providers who did not achieve Meaningful Use; this difference was not observed for counties with higher per capita incomes. Among counties with lower percentages of persons 65 years and older living in deep poverty, providers who achieved Meaningful Use had significantly lower

COVID-19 case rates than providers who did not achieve Meaningful Use (*P* = .02); this was not demonstrated in counties with higher percentages of persons 65 and older living in deep poverty. Among counties with lower unemployment rates, providers who achieved Meaningful Use had significantly lower COVID-19 case rates than providers who did not achieve Meaningful Use (*P* = .02); this was not demonstrated in those counties with higher unemployment rates.

## Discussion

### Principal findings COVID-19 death rates

We examined the association between achievement of Meaningful Use among Medicaid providers and cumulative reported county-wide COVID-19 death rates and found that Medicaid providers who achieved Meaningful Use were associated with significantly lower reported COVID-19 death rates than those who did not, regardless of provider and county population characteristics; results were similar for case fatality rates. We also found that SDoH were independently associated with covid death rates, similar to other findings [40]. This is the first study to our knowledge to address hypotheses concerning associations between COVID-19 reported outcomes and technology resources that reflect reporting of results from EHRs to public health agencies and facilitation of care coordination. Meaningful Use provides the mechanism to ease reporting to public health agencies; without this technology, it is plausible that reported case and death rates would be lower on average. In theory, public health and clinical data exchange provides the infrastructure for timely data reporting to policy makers. However, our results did not support this, and the actual use of EHR functions to report COVID-19 events effectively revealed the fragility of the data infrastructure as discussed by Hersh et. al. [41]. The second alternative hypothesis was that electronic and clinical data exchange improves quality of healthcare [12]. We provide evidence of more optimal patient outcomes (death and case fatality rate —measures of disease prevalence), associated with Medicaid providers who achieved Meaningful Use relative to those who had not; this supports providers' use of advanced functions of clinical data exchange [42], which may have also facilitated COVID-19 disease surveillance and risk stratification [43].

We assume that roughly half of the 3,716,747 Medicaid recipients in Florida are cared for by providers who did not achieve Meaningful Use (n = 1,858,374); we also assume that patients receiving care from providers who were classified subsequently as achieving or not achieving Meaningful Use, were of similar clinical status and had similar risk factors in conjunction with the pandemic. Then for an estimated .1% reduction in COVID-19-related deaths associated with achieving Meaningful Use, we expect 2631 excess COVID-19-related deaths in Florida alone that could possibly have been avoided had all Medicaid providers achieved Meaningful Use by November 2020. We acknowledge that the effect size is small. Because we are analyzing all data from a population of Florida's Medicaid providers and COVID-19 rates, a power analysis to support the size of a sample was not planned. Effect sizes have to be considered within the context of the prevalence of an attribute in a population, distribution of the exposure variable, and severity of the outcome; thus given the observational nature of this study, with death as the primary outcome, with approximately 50% of providers achieving Meaningful use, the small effect size is acceptable. The absence of a functional advanced EHR system universally implemented by Medicaid providers is associated with poorer patient outcomes. Unfortunately, we have no data to assess the assumption that patients of Medicaid providers who achieved Meaningful Use were of similar risk than those patients of providers who did not achieve Meaningful Use.

## Principal findings COVID-19 case rates

Regarding COVID-19 cases, the overall association between Meaningful Use among Medicaid providers and reported COVID-19 case rates was not significant. This suggests that increasing Meaningful Use may not reinforce public health measures to prevent the spread of disease. However, in post hoc analyses, the effect was modified by SDoH. Population health and its association with significant interactions between SDoH and Meaningful Use include the prevalence of respiratory diseases, and SDoH including personal income, unemployment, age, and poverty, indicating those living in deeper poverty and advanced age indicating those with less access to providers who achieved Meaningful Use are at significantly greater risk of COVID-19 infection. Overall, these findings are in accordance with those of previous studies [44, 45] and most recently with those of Palacio and Tamariz [46] who found that SDoH, such as poverty and unemployment, are accountable for up to 80% of poor health outcomes and 60% of deaths in the United States. Our finding that COVID-19 death rates were lower in persons ages 65 years and older living in deep poverty may be counterintuitive; however, if deaths were concentrated in long-term care facilities in which residents had to make financial investments to secure residence and the concentration of residents in these facilities was greater than alternative living situations, then the result is plausible.

While much has been reported on how SDoH influence clinical outcomes [47, 48, 54], our study also suggests that the population's socioeconomic composition may be associated with the provider's capacity to upgrade their practice's technology, particularly leveraging advanced EHR capabilities. These advanced EHR functions include the availability of real-time actionable patient data, health information exchange, clinical care coordination to achieve more optimal outcomes, decision support tools, and lastly, documentation of SDoH which relates to unintended adverse clinical outcomes prevalent in this vulnerable population [49].

Much has been reported on racial and ethnic disparities in US populations due to this pandemic [2]. Consistent with other studies, we found significantly greater rates of COVID-19 cases and deaths in the African American/Black population. However, we found substantially lower death rates in the Hispanic/LatinX populations. This appears to illustrate the "Hispanic Paradox," which refers to Hispanic and LatinX Americans tending to have health outcomes "paradoxically" comparable to, or in some cases better than those of their U.S. non-Hispanic White counterparts, even though Hispanics generally have lower average incomes and education [50, 51]. Our findings may be attributed partially to possible differences in age distributions and other demographic characteristics that could not be fully accounted for using existing clustered data. Regarding income, Figueroa et al. [52] found that lower median income was associated with higher COVID-19 death rates. Although this is confirmed in our study, we also show independently that greater % Per Capita Personal Income is associated with greater COVID-19 death rates. Our findings are consistent with those of Chin et al., which indicate significant intercounty variation in the distribution of household and community characteristics that affect the risk of infection and mortality from COVID-19 [53].

## Strengths and limitations

Although our results document the association between COVID-19 death and case fatality rates and achieving Meaningful Use accounting for provider and county characteristics, several limitations exist. First, this is an ecologic study with no individual-level data, limited county-level data, and no utilization data for providers. Thus, without patient-level data for age, sex, socioeconomic status, essential worker status, occupation, household size for individuals, and county-level data for local mask guidance and enforcement policies—key factors in evaluating morbidity and mortality, as well as no provider-level data concerning the extent to

which providers utilize EHR systems, the precision of estimates is potentially compromised. However, unlike other studies reported, we did have county-level data for chronic lung disease, and influenza and pneumonia. Second, although our definition of Meaningful Use allowed two years for the provider's achievement, some providers who enrolled in later years may have achieved Meaningful Use after the two years of observation (lead bias); we did not have complete data beyond 2018 and using any further data may have introduced bias. Third, our results pertain to Florida and may not be generalizable to those states that vary by demographic and socioeconomic characteristics, healthcare systems and public health enforcement. Furthermore, results pertain to Medicaid providers, but acknowledge that our COVID-19 case and death rates reflect all providers; however, our research question was focused solely on providers of care to the Medicaid population. Fourth, we acknowledge a disproportionately larger number of deaths occurring among nursing home patients. However, the HITECH Act excluded nursing homes and inpatient rehabilitation hospitals from the incentive program [54, 55]. Had nursing homes been eligible to participate in the program, we might expect their Meaningful Use death rate to be lower than that observed, thus widening the disparity in death rates between the Meaningful Use and non-Meaningful Use providers. Fifth, we did not have Federal Qualified Health Care designations, which may explain the difference between the provider's eligible volume criteria based on their own practice (solo) or reliance on their group practice. The designation of solo vs. group does not accurately reflect practice settings, given that some practitioners designated as 'solo' actually practice within a group setting; because these specific data were not available, interpretation of these associations would be misleading. Sixth, COVID-19 cases with no or mild symptoms may not have sought formal testing, and thus, contribute to underestimation of infection rates. Note: there were no COVID-testing sites or home testing kits by November 19, 2020. Since this time, several publications report lower than expected case rates of COVID-19 in Florida [56–61]. Additionally, there were only 55 dentists of the 3723 Medicaid providers who achieved Meaningful Use (1.5%), and thus reliability of findings may be of concern. Finally, the data were not available to identify whether or not the provider chose public health surveillance reporting to demonstrate Meaningful Use. However, as previously discussed, in 2014, only 6% of providers reported they chose Meaningful Use objectives to submit syndromic data to public health agencies; this further underscores the lack of readiness at that time to use the system for public health reporting.

## Conclusions

We found that achieving Meaningful Use was independently associated with lower COVID-19 death and case fatality rates, regardless of provider and county population characteristics.

Our findings also suggest that the association between Florida counties' public health outcomes and Meaningful Use achievement may have had less to do with using EHRs for reporting of clinical outcomes and more to do with using EHRs for coordination of care—a key measure of quality. The Florida Medicaid Promoting Interoperability Program in accordance with federal regulations, which incentivized Medicaid providers towards achieving Meaningful Use, has demonstrated success regarding both rates of adoption and clinical outcomes. Because the Program has ended in late 2021, we support continued EHR advanced use training and implementation in accordance with the goals of both HealthyPeople 2030 Health IT [62]. The Agency for Health Care Administration (AHCA) toward achieving desired technology advances in Florida, increasing value for providers and encouraging patient-centered care through improved access to relevant and meaningful information at the point-of-care [63].

This is in accordance with Sittig and Singh's report [64] regarding the COVID-19 pandemic, in that a government-mandated EHR system will help maximize scarce resources and

apply systematic healthcare data capture efforts to all patients while leveraging advances in technology to promote health equity.

## Supporting information

**S1 Data. Supporting raw datafile: S1_Data.xlsx.**
(XLSX)

**S1 Text. Output for interaction between age and deep poverty: S1_Text.doc.**
(DOCX)

## Acknowledgments

Amber Naranjo derived county level data from the Florida Department of Public Health.

## Author Contributions

**Conceptualization:** Katherine Freeman, Judith P. Monestime.

**Data curation:** Judith P. Monestime.

**Formal analysis:** Katherine Freeman.

**Methodology:** Katherine Freeman.

**Software:** Katherine Freeman.

**Writing – original draft:** Katherine Freeman, Judith P. Monestime.

**Writing – review & editing:** Katherine Freeman, Judith P. Monestime.

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
