## [Decision Letter · Decision Letter 0]

1 Mar 2022

PDIG-D-21-00134

Associations between Florida Counties’ COVID-19 Case and Death Rates and Meaningful Use Among Medicaid Providers: Cross-Sectional Ecologic Study

PLOS Digital Health

Dear Dr. Freeman,

Thank you for submitting your manuscript to PLOS Digital Health. After careful consideration, we feel that it has merit but does not fully meet PLOS Digital Health's publication criteria as it currently stands. Therefore, we invite you to submit a revised version of the manuscript that addresses the points raised during the review process.

The major issue to be addressed is the measured effect, whether it is true and whether if it is, it is significant. As I discuss, the authors are free to argue their view on this.

We look forward to receiving your revised manuscript.

Kind regards,

Dylan A Mordaunt, MB ChB, MPH, MHLM, FRACP, FAIDH

Academic Editor

PLOS Digital Health

Journal Requirements:

1. Please amend your Financial Disclosure statement. If you did not receive any funding for this study, please simply state: “The authors received no specific funding for this work.”

2. Please update the completed 'Competing Interests' statement. Please declare all competing interests beginning with the statement “I have read the journal's policy and the authors of this manuscript have the following competing interests:”

3. Please provide a complete Data Availability Statement in the submission form. If your research concerns only data provided within your submission, please write “All data are in the manuscript and/or supporting information files” as your Data Availability Statement.

4. We have noticed that you have uploaded raw data files as supporting information but you have not included them in the list of legends. Please add a citation of this SI data files into the list of legends of your supporting information files.

Additional Editor Comments (if provided):

Thank you for your submission. With regards to the criteria for publication:

- The manuscript appears to present the results of original research with a degree of novelty in terms of the focus.

- High importance and broad interest to community of researchers, engineers and clinicians working in the field of digital health- digital health is a fairly broad remit, ranging from development of technology through to impact of implementation. I think this does fit within the wider remit of digital health, with the methods being somewhat outside what is typically presented in digital health.

- High methodological rigor and ethical standards- reviewer 3 has raised concerns about whether there are real differences and therefore whether the authors aren't over-interpreting the data. I think the authors either need to carefully address this or otherwise argue why it should be addressed following publication. I don't agree with the reviewer's view that this should result in publication since the matter could be addressed either during peer review or following publication.

- Substantial evidence for its conclusions- this goes to the issue above.

- Clearly outlined utility and accessibility for the broader community.

- Follow appropriate standards and practice of open science.

Reviewers' comments:

Reviewer's Responses to Questions

**Comments to the Author**

1. Does this manuscript meet PLOS Digital Health’s publication criteria? Is the manuscript technically sound, and do the data support the conclusions? The manuscript must describe methodologically and ethically rigorous research with conclusions that are appropriately drawn based on the data presented.

Reviewer #1: Yes

Reviewer #2: Yes

Reviewer #3: Partly

2. Has the statistical analysis been performed appropriately and rigorously?

Reviewer #1: Yes

Reviewer #2: Yes

Reviewer #3: N/A

Reviewer #4: Yes

3. Have the authors made all data underlying the findings in their manuscript fully available (please refer to the Data Availability Statement at the start of the manuscript PDF file)?

Reviewer #1: Yes

Reviewer #2: Yes

Reviewer #3: Yes

Reviewer #4: Yes

4. Is the manuscript presented in an intelligible fashion and written in standard English?

Reviewer #1: Yes

Reviewer #2: Yes

Reviewer #3: Yes

Reviewer #4: Yes

5. Review Comments to the Author

Reviewer #1: This is overall a well-written manuscript presenting the " Associations between Florida Counties’ COVID-19 Case and Death Rates and Meaningful Use Among Medicaid Providers: Cross-Sectional Ecologic Study".

- It clearly states the problem to be solved and specified the nature of the knowledge sought in the study.

- Manuscript is well organized and flows well. The language is unbiased and the length is appropriate. 

- The writing give substantive evidence of the authors' familiarity with the subject, literature and concepts presented in the study. References are scholarly, current and appropriate

- The content is innovative and builds and advances body of knowledge of Ecologic Studies / Public Health Data and Health Informatics regarding to the associations between Florida Counties’ COVID-19 Case and Death Rates and Meaningful Use Among Medicaid Providers:

- . The design of the research is sound. The description of the methodology, findings and relevancy is identifiable, precise and accurate.

- The Data analysis is clear, relevant and thorough to support the conclusion.

Reviewer #2: The authors analyse an interesting and novel aspect of the COVID-19 pandemic, and they took a lot of variables into consideration. An extensive statistical analysis is also provided.

There is one thing which might worth mentioning: the incident rates are based on the reported case numbers, but not all COVID-19 cases are detected. People with no or mild symptoms might not report in the health care system, and limitation of testing kits or human work force might result in underestimating the case numbers. Do you have any information regarding these issues in the analysed regions, especially if there are any differences among the provides which might affect your conclusions?

The manuscript contains some minor editing errors, including some typos on page 5 and an "[Error!

Bookmark not defined.]" on page 12.

Reviewer #3: The authors present a study of COVID-19 death, case and case-fatality rates (CFR) across Florida counties. They report that death rates and CFR, but not case rates, were associated with Meaningful Use of EHRs achieved by Medicaid providers. The authors claim that their findings have implications for policymaking, giving evidence that government support to adopt EHRs among Medicaid providers has beneficial effects.

Major points:

1. While the authors report significant differences, the effect sizes are very small and differences are well within standard deviations. This raises the suspicion that the small p-values are simply due to the large sample size, and not a true underlying difference. The authors should confirm that the reported p-values are correct and should comment on this issue.

2. In particular, given the small effect sizes, it does not seem justified to derive from this a strong case for policies regarding EHR adoption, especially since the authors could not confirm that patients of Medicaid providers who achieved Meaningful Use were of similar risk to patients of providers who did not. The authors need to at least provide additional justification for why they believe their data provides good evidence for extending government support beyond 2021.

3. The authors performed a large number of comparisons and, as is to be expected, found some significant associations. However, in addition to small effect sizes (point 1), any real causative associations would also need to be plausible. For example, it is difficult to imagine a plausible mechanism by which EHR adoption by dentists could be causally related to COVID-19 case rates. Similarly, why would COVID-19 death rates be lower in counties with a higher concentration of persons ages 65 years and older living in deep poverty, but at the same time positively correlate with higher rates of those living in poverty? The authors need to provide their rationale behind not excluding such associations as unsound.

4. Even if there is a real difference in COVID-19 death rates and CFR among Medicaid providers who did and did not achieve Meaningful Use, the authors do not provide evidence that EHR adoption was the causal factor. It could be that providers who achieved Meaningful Use offer better care quite independent of EHR adoption. Again, they would need to give additional justifications for deriving policy implications from their results.

Minor points:

1. The authors correctly state that Meaningful Use was captured before the COVID-19 pandemic, and therefore case and death rates are not confounded by it. However, their Methods suggest that the last follow-up took place in 2018. It would strengthen their study if they could include more recent data showing that those providers who sustained Meaningful Use in 2018 were still doing so during the COVID-19 pandemic.

2. A reference is broken on p.12.

Reviewer #4: This work addresses the impact and differences between Medicaid providers who did and did not 

implement electronic health records and achieve Meaningful Use as it relates to Florida county-level cumulative COVID-19 death, case and case fatality rates, taking into consideration the county level demographics, socio-economic and clinical markers, and healthcare environment. 

This work had several findings. Below are some of the findings:

(a) The findings from this work show that there were significant differences in the cumulative incidence rates of COVID-19 deaths and case fatality rates between the 5025 Medicaid providers not achieving Meaningful Use and the 3723 that achieved Meaningful Use.

(b) County level characteristics such as lower median household income, greater concentrations of persons of African American or Black race, higher unemployment, and higher concentrations of those living in poverty and without health insurance, were independently associated with increased COVID1-9 death rates and case fatality rates.

(c) The association between Florida counties public health outcomes and Meaningful Use achievement most likely had less to do with using electronic health records for reporting of clinical outcomes and had more to do with using electronic health records for coordination of care

My comments

I enjoyed reading this paper. It was well written and the analysis and results were well reported and I thought the discussions section was detailed and addressed the analysis, results, and findings from this work.

I think this paper should be accepted. It is my opinion that the research community will benefit from this paper being published. Also, this work and its findings could potentially encourage other researchers to conduct similar work in various states across the United States.

Below is a minor comment

In this work, one of the county-level characteristics associated with lower COVID-19 death rates was a higher concentration of person’s ages 65 and older living in deep poverty. Did the authors conduct further analyses to gain insight about this finding?

6. PLOS authors have the option to publish the peer review history of their article (what does this mean?). If published, this will include your full peer review and any attached files.

**Do you want your identity to be public for this peer review?** For information about this choice, including consent withdrawal, please see our Privacy Policy.

Reviewer #1: Yes: Dr. Audrey Blackwood, RHIA

Reviewer #2: No

Reviewer #3: No

Reviewer #4: No

**Comments to the Author**

1. Does this manuscript meet PLOS Digital Health’s publication criteria? Is the manuscript technically sound, and do the data support the conclusions? The manuscript must describe methodologically and ethically rigorous research with conclusions that are appropriately drawn based on the data presented.

Reviewer #4: Yes

---

## [Editor Report · Decision Letter 1]

20 Apr 2022

Associations between Florida Counties’ COVID-19 Case and Death Rates and Meaningful Use Among Medicaid Providers: Cross-Sectional Ecologic Study

PDIG-D-21-00134R1

Dear Dr. Freeman,

We are pleased to inform you that your manuscript 'Associations between Florida Counties’ COVID-19 Case and Death Rates and Meaningful Use Among Medicaid Providers: Cross-Sectional Ecologic Study' has been provisionally accepted for publication in PLOS Digital Health.

Best regards,

Dylan A Mordaunt

Academic Editor

PLOS Digital Health

Thank you for your resubmission. This now meets the criteria for publication.